# Exploring Prior Knowledge from Human Mobility Patterns for POI Recommendation

Jingbo Song [1], Qiuhua Yi [2], Haoran Gao [3], Buyu Wang [4] and Xiangjie Kong [2,*]

1 School of Arts, Tourism College of Zhejiang, Hangzhou 311231, China; janet@tourzj.edu.cn
2 College of Computer Science and Technology, Zhejiang University of Technology, Hangzhou 310023, China; qiuhuayi@outlook.com
3 School of Software, Dalian University of Technology, Dalian 116620, China; damiangao96@gmail.com
4 College of Computer and Information Engineering, Inner Mongolia Agricultural University, Hohhot 010018, China; bywang08@imau.edu.cn
* Correspondence: xjkong@ieee.org; Tel.: +86-158-4090-1926

**Abstract:** Point of interest (POI) recommendation is an important task in location-based social networks. It plays a critical role in smart tourism and makes it more likely for tourists to have personalized travel experiences. However, most current recommendation methods are based on learning the users' check-in history and the users' relationship network in the social network to make recommendations.Therefore, urban crowds' regular travel patterns cannot be effectively utilized. In this paper, we propose a POI recommendation algorithm (HMRec) based on prior knowledge of human mobility patterns to solve this problem. Specifically, we propose the Human Mobility Pattern Extraction (HMPE) framework, which utilizes graph neural networks as extractors for human mobility patterns. The framework incorporates attention mechanisms to capture spatio-temporal information in urban traffic patterns. HMPE employs downstream tasks and design upsampling modules to reconstruct representation vectors for task objectives, enabling end-to-end training of the framework and obtaining pre-trained parameters for the human mobility pattern extractor. Furthermore, we introduce the Human Mobility Recommendation (HMRec) algorithm, which improves feature cross-interactions in the breadth model and incorporates prior knowledge of human patterns. This ensures that the recommendation results align more closely with human travel patterns in urban environments. Comparative experiments conducted on the Foursquare dataset demonstrate that HMRec outperforms baseline models with an average performance improvement of approximately 3%. Finally, we discuss existing challenges and future research directions, including approaches to address the issue of data sparsity.

**Keywords:** POI recommendation; human mobility pattern; graph neural network





## 1. Introduction

Undoubtedly, smart tourism is an important component of smart cities. As tourist destinations continue to develop and change, people's travel choices are becoming more diverse and complex. Smart tourism aims to intelligently connect and manage cities and tourism resources using technologies such as the Internet of Things (IoT), big data, and artificial intelligence in order to provide personalized tourism services to travelers [1]. In the context of deep learning technology, Point of Interest (POI) recommendation plays a crucial role in smart tourism. It can extract preference information from users' historical travel data and provide them with greater convenience and satisfaction in their travel experiences. POI recommendation can also alleviate the current situation of over-tourism [2], reducing the burden and environmental pressure of tourism destinations.

On the other hand, the Internet of Things (IoT), as a network infrastructure, collects various needed information in real-time through various technologies, realizes the ubiquitous connection between things and other things and between things and people, and

achieves intelligent perception of objects and processes [3,4]. Through sensors and devices in the Internet of Things (IoT), environmental information, foot traffic, traffic conditions, and other data surrounding Points of Interest (POIs) can be collected and monitored. These dynamic and heterogeneous data can be used for real-time monitoring, analysis, and prediction of POIs, thereby providing more accurate recommendations and services.

Due to the widespread use of mobile positioning devices (smartphones) and the rapid development of Location-Based Social Networks (LBSNs) [5], users can share location data by checking in at POIs on social networks [6]. Through LBSNs, users can record and share their activities and experiences at different POIs, generating a large number of POI-related data [7]. These POI-related data include users' interests and behavioral information at specific locations, which can reflect the preferences of user groups. These data can be used to discover human mobility patterns and provide important reference for POI recommendation systems. POI recommendation is an important way to assist users in exploring the surrounding environment to improve user experience. The POI recommendation algorithm provides users with personalized recommendation services by learning user historical data. At present, the mainstream POI recommendation system focuses on learning users' preferences from historical social network data to make recommendations [8]. However, the results obtained in this way may be counter-intuitive, such as recommending a downtown cafe on a non-weeknight. Therefore, POI recommendation should combine prior knowledge of human mobility patterns [9].

In order to efficiently explore prior knowledge from human mobility patterns, various data in cities (traffic trajectory data, map data, location data, social media data, consumption data, etc.) should be effectively utilized. At present, understanding how to effectively represent the relationship between multi-source heterogeneous data in cities and how to extract and analyze the features of multi-source heterogeneous data relationships are still important and urgent problems to be solved in the research on human mobility patterns. In addition, studies purely aimed at extracting human mobility patterns are rare. Human mobility patterns are usually an intermediate result of completing a specific downstream task. Understanding how to select downstream tasks and verify the quality of the obtained human mobility patterns is also a problem investigated in this paper.

To address these problems, we propose a POI recommendation method based on prior knowledge of human mobility patterns. This method makes full use of the prior knowledge of human mobility patterns to solve the difficult problem of choice caused by a large amount of promotional information flooded in social networks. The contributions of this paper are summarized as follows:

(1) We propose the Human Mobility Pattern Extraction (HMPE) architecture to learn human mobility patterns and extract their representation.
(2) We design a traffic congestion prediction task and conduct experiments on a real taxi dataset from New York, which verifies the feasibility of using an end-to-end architecture to learn human mobility patterns.
(3) We propose a Point of Interest recommendation algorithm called Human Mobility Recommendation (HMRec), which incorporates prior knowledge of human mobility patterns.
(4) We conduct a comparative experiment on the Foursquare dataset, and the experimental results validate the effectiveness of prior knowledge of human mobility patterns.

The rest of this paper is organized as follows. Section 2 introduces the research background and significance of the POI recommendation system and then describes the research status of human mobility pattern discovery and POI recommendation at home and abroad. Section 3 presents the introduction of related theories, focusing on the discovery of human mobility patterns, the POI recommendation algorithm, and related theories and technologies involved. Section 4 provides the experimental design and analysis of results. Section 5 summarizes the specific contributions and implications of this paper and discusses possible future research directions.

## 2. Related Work

In this section, we introduce the latest progress in POI recommendation and human mobility patterns.

### 2.1. POI Recommendation

The development of POI recommendation can be attributed to the continuous innovation of data fusion methods, from geographical location information to users' social relations to the spatiotemporal patterns of cities; more and more key elements have been discovered by researchers as new features of recommendation algorithms [10].

The early POI recommendation algorithm uses the user's check-in frequency to make recommendations [11]. Berjani et al. [12] used regularized matrix factorization to provide personalized recommendations to users in social networks. However, the raw data values recommended by POI have a very wide range. In addition, the sparsity of the user check-in matrix is much larger than that of the item-rating matrix. For example, the sparsity of the Netflix dataset is 99%, while the sparsity of Gowalla is $2.08 \times 10^{-4}$. These situations facilitate POI recommendations for data fusion [13]. In the data fusion method, the earliest introduction of POI recommendation is the geographic location information. For POI recommendation systems, the first law of geography is shown as follows: users are more interested in POIs that are geographically closer to them, and they will have higher interest in POIs near the POIs they are interested in. Si et al. [14] proposed an adaptive POI recommendation method combining users' activities and spatial features, which can operate adaptively according to users' activities. Liu et al. [15] proposed the Geo-ALM model, which utilizes adversarial learning with geographic information to fuse geographic features and generative adversarial networks. In LBSNs, some POI recommendation works utilized social relationship networks to improve recommendation quality [7,16]. Zhang et al. [17] formed a three-layer model of LBSN with multi-label, social, and geographic influences and finally integrated multi-dimensional information into a matrix factorization framework. Zhang et al. [18] proposed a contextual graph attention model, and Wang et al. [19] used a graph-enhanced spatial-temporal network for POI recommendation.

However, research on POI recommendations is still in its preliminary stage. Although the existing work takes into account the influence of space, social interaction, time, and other factors on the recommendation effect of points of interest, the recommendation target tends to fit the historical check-in behaviors of individual users who are unable to learn the travel rules of people in the city and finally obtains a recommendation result that does not conform to common sense.

### 2.2. Human Mobility Pattern

In order to efficiently mine human mobility patterns and introduce them into the point of interest recommendation task, all kinds of data in the city should be effectively utilized [20,21]. Therefore, we need to solve two problems: urban multivariate data fusion and human mobility pattern mining.

For urban multivariate data fusion, H. Zhang et al. [22] used taxi data, shared bike data, and road network data to detect abnormal areas in stages. They used different data at different stages of the task process to achieve the purpose of data fusion. X. Liu et al. [23] proposed a deep convolutional auto-encoder architecture that enables the encoding layer to learn the features of different data. Some work [24–26] used the idea of collaborative filtering to establish the relationship between different data sources for data fusion. The fusion of multivariate data requires the algorithm to have strong scalability, but there is no general method to meet the needs.

The human mobility pattern itself is also a spatio-temporal pattern, and spatio-temporal pattern mining directly affects the quality of POI recommendations. In order to adapt to different spatio-temporal networks, Li et al. [27] proposed an efficient spatio-temporal neural structure search method to try to automatically construct a general neural network suitable for different spatio-temporal prediction tasks in cities. Due to the lack

of semantics in the spatio-temporal data features generated using deep neural networks, Wang et al. [28] proposed a new deep learning framework based on particle swarm optimization. It can be seen that the research on human mobility patterns is often related to people's travel behaviors and trajectories, and the pros and cons of human mobility pattern mining results often require downstream tasks for verification and support.

Compared to the above works, we propose the architecture of human mobility pattern extraction and then design the end-to-end architecture to learn the characteristics of human mobility patterns, and finally apply the prior knowledge of human mobility patterns to POI recommendations.

## 3. Methodology

### 3.1. Human Mobility Pattern Discovery

#### 3.1.1. Spatio-Temporal Graph

Inspired by the widely used data structure of graphs, the spatio-temporal graph data structure proposed in this paper can more accurately represent the spatio-temporal properties of cities [29]. As long as the data can be structured using graphs, spatio-temporal graphs can generally be adapted [30]. We formulate the spatio-temporal graph as $G_{seq} = \{G_0, G_1, G_2, \dots\}$, and it can be seen that it consists of a sequence of spatial graphs aggregated by multiple time slices. The spatial graph is defined as $G_t(V_t, E_t, A_t)$, where $V_t$ represents the set of all nodes in the time slice, $E_t$ represents the set of edges in the time slice, and $A_t$ represents the attribute set of the nodes in the time slice. We adopt spatiotemporal graphs to structurally represent the human mobility pattern of cities and the functional properties of urban areas.

#### 3.1.2. Human Mobility Pattern Discovery Architecture

The human mobility patterns discovery framework uses the prediction of downstream tasks to pre-train the human-mobility-pattern-extraction module, so that the human-mobility-pattern-extraction module obtains the ability to represent the spatio-temporal pattern under the specific city data distribution. At the same time, the prediction results of the downstream tasks themselves can help to test the representation effect of human mobility patterns. The overall architecture is shown in Figure 1. The human mobility pattern-extraction framework has three main parts, the human-mobility-pattern-extraction module, the encoding and decoding module based on the multi-head attention mechanism, and the upsampling module adapted to the downstream task. The three modules form a unified end-to-end model, and the resulting intermediate hidden layer vectors contain the spatio-temporal contextual relations of the human mobility pattern.

The human-mobility-pattern-extraction module is composed of a graph convolutional neural network with shared weights and a feedforward neural network [19]. Given a graph $G(V, E)$, where $V$ and $E$ are the sets of points and edges of the graph, respectively, with the point set size $n = |V|$ and the edge set size $m = |E|$. The adjacency matrix of graph $G$ is $A \in \mathbb{R}^{n \times n}$. Assuming that there are connected edges between nodes $i$ and $j$, then $A_{ij} = 1$; otherwise $A_{ij} = 0$. The degree matrix of graph $G$ can be obtained as $\boldsymbol{D} = diag(d_1, d_2, \dots, d_n)$, where $d_i$ is the degree of node $i$ in graph $G$. The Laplacian matrix of the graph is $\boldsymbol{L} = \boldsymbol{D} - \boldsymbol{A}$. We perform an eigenvalue decomposition of $\boldsymbol{L}$ to obtain $\boldsymbol{L} = \boldsymbol{U}\boldsymbol{\Lambda}\boldsymbol{U}^T$. Defining the graph signal $\boldsymbol{x} = [x_1, x_2, \dots, x_n]^T$, for any graph signal $x$, we can obtain the following formula:

$$
\begin{aligned}
\boldsymbol{x}^T \boldsymbol{L} \boldsymbol{x} &= \boldsymbol{x}^T \boldsymbol{D} \boldsymbol{x} - \boldsymbol{x}^T \boldsymbol{A} \boldsymbol{x} \\
&= \sum_i d_i x_i^2 - \sum_{(i,j) \in E} A_{ij} x_i x_j \\
&= \sum_{(i,j) \in E} (x_i - x_j)^2
\end{aligned}
\tag{1}
$$

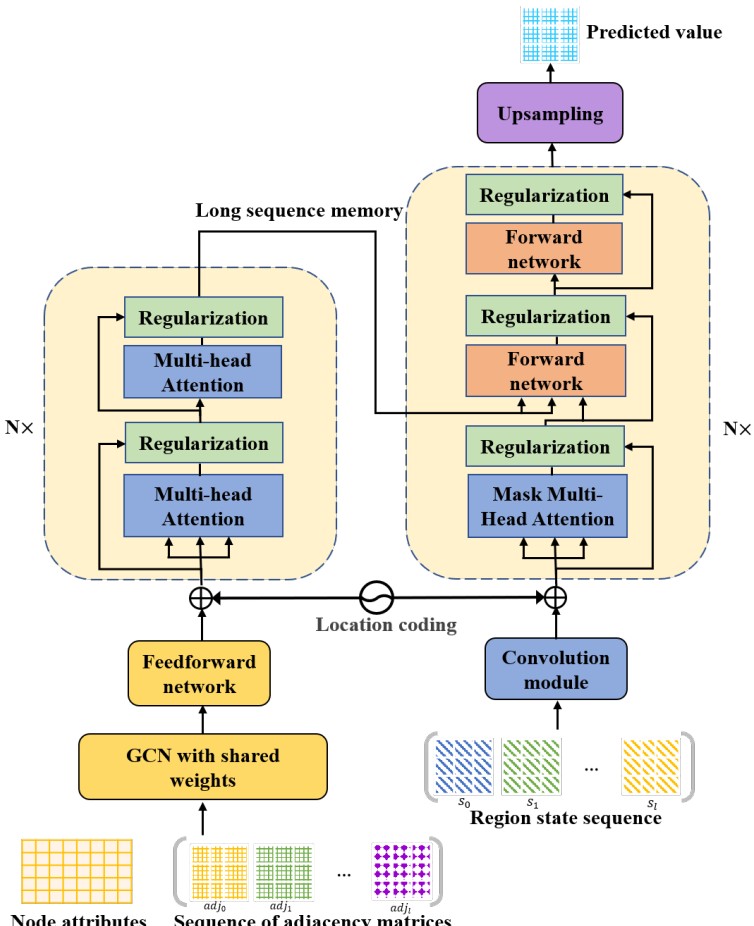

**Figure 1.** Human-mobility-pattern-discovery architecture. The human mobility pattern extraction module consists of a convolutional neural network with a shared weight graph and a feedforward neural network. The encoding and decoding module based on the multi-head attention mechanism refers to the encoding and decoding structure of Transformer [31]. The upsampling module uses a deconvolutional network.

So far, we can derive the corresponding inverse transform $\boldsymbol{x} = \boldsymbol{U}\hat{\boldsymbol{x}}$ of the Fourier transform $\hat{\boldsymbol{x}} = \boldsymbol{U}^T\boldsymbol{x}$ of the graph signal. The convolution form of the graph signal $x$ and the convolution kernel $h$ on graph $G$ can be obtained, as shown in Formula (2).

$$\boldsymbol{x} * \boldsymbol{h} = \boldsymbol{U} \cdot diag(\hat{\boldsymbol{h}}) \cdot \boldsymbol{U}^T\boldsymbol{x} \tag{2}$$

We replace $diag(\hat{\boldsymbol{h}})$ above with $\sum_{k=0}^{K} \alpha_k \Lambda^k$, where $\{\alpha_k\}_{k=0}^{K}$ is a learnable parameter. So far, we have obtained a graph convolutional neural network model that does not require feature decomposition and has a computational complexity of $O(n)$:

$$\begin{aligned} y &= \sigma(\boldsymbol{U} \cdot (\sum_{k=0}^{K} \alpha_k \Lambda^k) \cdot \boldsymbol{U}^T\boldsymbol{x}) \\ &= \sigma(\sum_{k=0}^{K} \alpha_k \boldsymbol{L}^k\boldsymbol{x}) \end{aligned} \tag{3}$$

In this module, the graph signal refers to the functional attribute distribution vector of the urban area, and the adjacency matrix is the spatio-temporal graph of the flow between urban areas. We set the matrix after rasterization of the city to have $r \times r$ areas, the attribute vector of each area to have $d_{fi}$ dimensions, and the area representation dimension after

graph convolution to be $d_{fo}$. The graph neural network framework used in this module can be simplified as follows:

$$Y = \sigma(\hat{D}^{-1}\hat{A}XW) \tag{4}$$

where $X \in \mathbb{R}^{r^2 \times d_{fi}}$ is the city function attribute matrix, $A \in \mathbb{R}^{r^2 \times r^2}$ is the traffic adjacency matrix, $\hat{A} = A + I$, $\hat{D}$ is the degree matrix corresponding to $\hat{A}$, and $W \in \mathbb{R}^{d_{fi} \times d_{fo}}$ is the network parameter to be learned. $XW$ linearly changes the feature vector of the node, $\hat{A}XW$ propagates the transformed node feature to neighbors, and $\hat{D}^{-1}\hat{A}XW$ normalizes the feature received by the node.

In this paper, the graph convolutional neural network [32,33] to which each time slice belongs shares the parameter weight, and only one graph neural network is trained and maintained. For the input backpropagation gradients of different time slices, we use the sum-average processing. The human mobility pattern extraction strategy learned by the graph neural network is applicable to all time slices [34].

Since the input data of the coding module are a sequence composed of the representation vectors of urban human mobility patterns under a single time slice instead of the whole graph representation, we use the feedforward neural network [35] module to realize the function of the ReadOut module in DGI [36] and obtain the hidden vector representation of the whole graph.

In the encoder–decoder module based on the multi-head attention mechanism, we refer to the encoder–decoder structure of Transformer [37]. We use a sequence of representation vectors of human mobility patterns as input representations and target data representations of downstream tasks as output representations. The attention mechanism can be represented by the following formula:

$$Attention(Q, K, V) = softmax\left(\frac{QK^T}{\sqrt{d_k}}\right)V \tag{5}$$

We set the original attention module to be replicated $h$ times, i.e., $h$ head attention. We input the same data to the multi-head attention module to obtain $h$ single-head attention results. We then spliced the multi-head results and transformed them through a linear layer to obtain the fused feature output [38]. The multi-head attention formula is as follows:

$$MultiHead(Q, K, V) = Concat(head_1, head_2, \dots, head_h)W^o \tag{6}$$

We used the human-mobility-pattern-extraction module to replace the original list of embedding in the transformer. The representation is directly extracted from the original data, and the output module is replaced by the downstream-task feature-extraction module. The upsampling module is used to restore the predicted representation vector to the same structure as the real data. In this paper, we use the deconvolution network as the upsampling module and generate a map of the urban traffic congestion situation from the predicted representation vector $\hat{emb} \in \mathbb{R}^d$, denoted as $\hat{S} \in \mathbb{R}^{r \times r}$. We calculate the average speed of $r \times r$ nodes in the time period, and the average speed of each node in each time slice can be regarded as a map snapshot of urban traffic congestion. We denote the snapshot of the congested state of the city-market traffic corresponding to the time slice $t$ as $S_t \in \mathbb{R}^{r \times r}$. We thereby formulate the task of urban traffic congestion state assessment. Given the spatio-temporal graph sequence of traffic flow as $G_{seq} = \{G_1, G_2, \dots, G_t\}$, the corresponding function distribution of each region as $A \in \mathbb{R}^{r^2 \times d_{fi}}$, the sequence of urban traffic congestion state as $A \in \mathbb{R}^{r^2 \times d_{fi}}$, and the prediction of urban traffic congestion state in time slice $t$ as $S_t$, let the output result of the overall framework be $\hat{S}_t$, and expect $\hat{S}_t$ to be as close as possible to $S_t$. Then, the optimization objective of the overall framework is

$$L = \|\hat{S}_t - S_t\| \tag{7}$$

In order to adapt to the data characteristics, we used the convolutional neural network in the downstream task feature extraction module of this paper and the deconvolutional neural network in the upsampling module.

### 3.2. POI Recommendation Algorithm Based on Human Mobility Patterns

In a typical location-based social network (LBSNs), the POI recommendation system has a set of $N$ users $U = \{u_1, u_2, \ldots, u_N\}$ and a set of $M$ geographic locations $L = \{l_1, l_2, \ldots, l_N\}$, also known as the set of interest points. The set of interest points accessed by user $u$ is represented by $L_u$. Each location $l_i$ is geocoded with *<longitude, latitude>*. Generally, we convert the user's check-in information into the user-POI check-in frequency matrix $C$. Each entry $c_{ui}$ of $C$ represents the check-in frequency of user $u$ at place $i$, and the check-in frequency reflects the user's preference for different points of interest. Typically, the user visits only a small number of locations, so the matrix $C$ is very sparse. We retain the weight of the human mobility pattern extractor model and separate it out separately. The result is used as the part of urban human mobility feature extraction of the recommendation algorithm [39]. The HMRec algorithm borrows from the Wide&Deep model [40], so they have similar time complexity, approximately $O(d + Ln)$, where $d$ represents the dimensionality of the input features, $L$ is the number of layers in the model, and $n$ is the number of neurons per layer. The specific structure of the model is shown in Figure 2.

The HMRec model utilizes multiple cross layers for feature crossing [41]. If the output vector of the $l$-th Cross layer is $x_l$, then the output of the $l + 1$-th layer is

$$x_{l+1} = x_0 x_l^T W_l + b_l + x_l \tag{8}$$

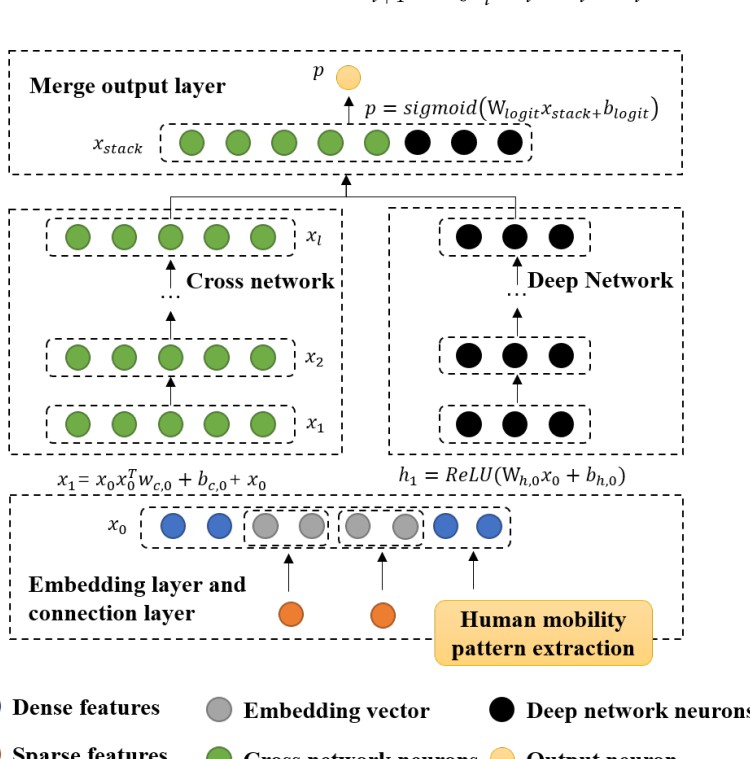

**Figure 2.** HMRec algorithm structure. The input of the deep model is a full amount of feature vectors, covering the geographic location features, time features, category features of POIs, and ID class features of users and POIs. The inputs to the Cross model are user ID, POI ID, and POI category.

The second-order operation of the cross layer is similar to the outer product operation in the PNN model, on top of which we add the weights $w_l$ of the outer product operation, as well as the input $x_l$ and bias $b_l$. It can be seen that the cumulative effect of the superposition of the Cross layers on the parameter quantity is relatively slow, and each layer only adds an

$n$-dimensional weight vector $\mathbf{w}_l$, where $n$ is the dimension of the input vector. In addition, the original input vector $\mathbf{x}_l$ is preserved at each layer so that the variation between input and output is small. We retain the weight of the human mobility pattern extractor model and separate it out separately, and it is used as part of the urban human mobility feature extraction of the recommendation algorithm [42]. The overall algorithm flow is shown in Algorithm 1.

---

**Algorithm 1** HMRec Algorithm

---

**Input:** Features of urban human mobility pattern $f_h$, point of interest $ID_i$, user $ID_j$, dense features $f_d^1, f_d^2, \ldots, f_d^m$, discrete features $f_s^1, f_s^2, \ldots, f_s^n$
**Output:** The predicted score $p$ of user $j$ checking in at POI $i$
    **Part I: Cross Layer**
  1: transform the discrete features into the dense features: $f_d^{m+k} = Embedding(f_s^k)$
  2: splice dense features: $f_d = Cat(f_d^1, f_d^2, \ldots, f_d^{m+n}, f_h)$
  3: using deep networks to fuse dense features: $h_0 = ReLU(W_0 f_d + b_0)$
  4: get dense features: $h_{l+1} = ReLU(W_l h_l + b_l)$
    **Part II: Deep Layer**
  5: discrete feature multi-layer high-order intersection: $x_{l+1} = x_0 x_l^T W_l + b_l + x_l$
  6: connect the Deep network and cross network and output: $x_c = Cat(h, x)$
  7: **return** return predicted score $p = sigmoid(W x_c + b)$

---

## 4. Results and Discussion

### 4.1. Urban Traffic Congestion Status Assessment Task

In this paper, in order to ensure that the spatio-temporal pattern extraction framework can learn urban spatio-temporal features and at the same time evaluate the learning effect of this feature, we design urban congestion situation prediction as a downstream task of the extraction framework. This is also a general practice for evaluating the performance of vector representations in the hidden middle layers of an encoding–decoding framework [43].

We used yellow taxi data and POI data from May and June 2012 in New York City to conduct a comparative experiment. The experiment selected the spatial range of longitude from $-74.0108$ to $-73.9600$ and latitude from $40.7333$ to $40.7700$ in New York City. Within the time and space selected for the experiment, the number of taxi trajectory data was about 9 million, the number of point-of-interest data was 1612, and there were 18 types of points of interest.

In this study, we designed ablation experiments to verify the effectiveness of the HMPE algorithm. In this paper, a method similar to unsupervised learning verification was adopted to evaluate the learning effect of the model on target hidden vectors using the performance results of the model in downstream tasks. The HMPE algorithm framework proposed in this paper mainly needs to verify two parts: the human mobility pattern extraction module and the timing feature extraction module. Therefore, the control part of the ablation experiment was set as the representative algorithm of each part.

For the human mobility pattern extraction module, there are two parts in the framework; one is the graph representation algorithm with attributes, and the other is ReadOut. Consider the Node2Vec algorithm [44] (Scalable Network Feature Learning) as a representative of a class of algorithms for attribute-free, purely structured representations. We used ANRL [45] (Deep-Neural-Network-Based Attribute Network Representation Learning) to compare the differences between attribute-based graph-convolution algorithms and attribute-based random–walk algorithms. In the ReadOut section, we used a convolutional neural network to verify the difference between city-wide global view (MLP) and local view (CNN [46]) in urban spatio-temporal scenes. For the time series feature-extraction module, the GRU [47] in the long-term and short-term memory network was used as the representative algorithm to compare the difference between learning only a single

long-term and short-term memory vector and using the attention mechanism to treat the features of each time period differently in the urban spatio-temporal scene [48,49].

In the experiment, we set 200 epochs, the test set ratio was set to 0.3, the training batch size was set to four, the side length of the city raster division was set to 16, and the time series length was set to 24. In addition, in order to ensure the singleness of the ablation experiment variables, we limited the size of the output vector of the human mobility feature module to 32 and set the number of channels to eight. The computational focus of the experiment lies primarily in the encoding and decoding modules of the multi-head attention mechanism. The architecture of this module is inspired by the encoding and decoding structure of the transformer, so the time complexity is similar. The time complexity of the transformer model primarily depends on the computational complexity of its self-attention mechanism and feed-forward neural network. The time complexity per layer is approximately $O(n^2 d)$, where $n$ represents the sequence length and $d$ represents the representation dimension. We use Mean Absolute Percentage Error (MAPE) and Rooted Mean Square Error (RMSE) as the evaluation indicators of the framing effect.

The experimental results are shown in Table 1, and it can be seen that the performance of the HMPE framework proposed in this paper is better than other baseline models in terms of RMSE and MAPE. This shows that the HMPE framework performs as expected, effectively extracting human mobility patterns in cities. Compared with the structure using Node2Vec, HMPE makes better use of the features of urban functional areas. Compared with using CNN as the structure of ReadOut, HMPE uses MLP to achieve a global receptive field instead of a local receptive field and achieves better results. Since the two structures of Node2Vec and ANRL are staged models, the results obtained by the end-to-end training of HMPE are obviously superior. Note that the performance of GCN-MLP-GRU is also quite good, which to some extent shows that a good representation of urban spatiotemporal features is very helpful for downstream tasks.

**Table 1.** Results of urban traffic congestion prediction.

| Evaluation Indicator | GCN-CNN-Trans | N2V-MLP-Trans | ANRL-MLP-Trans | GCN-MLP-GRU | HMPE |
| --- | --- | --- | --- | --- | --- |
| **RMSE** | 0.1693 | 0.1517 | 0.1807 | 0.1473 | 0.1466 |
| **MAPE** | 34.70% | 28.39% | 44.71% | 26.06% | 25.65% |

### 4.2. POI Recommendation Based on Human Mobility Pattern

We selected the travel data and check-in data of New York City in May and June 2012 for the experiment. The user check-in data came from the Foursquare dataset publicly available on Kaggle, which contains 227,428 user check-in data. We aggregated the information on the check-in time according to the hour, and obtained 1464 time slices of the check-in data. Arranging the check-in data in chronological order, we took the first 70% of the data as the training set and removed the points of interest that users have checked in to from the remaining 30% data to obtain the test set.

Comparative experimental baseline models include the deep factorization machine (DeepFM) [50], the representational multi-layer perceptron (EmbeddingMLP), the deep-learning-based collaborative filtering model (NeuralCF) [51], the Wide&Deep model, and the Twin-tower model [52]. In the comparison experiment on the POI recommendation model, this paper used the accuracy rate (ACC) and two AUC values (ROC and PR) as the experimental indicators.

The experimental results are shown in Table 2. It can be seen that the HMRec algorithm proposed in this paper achieved results that exceed all baseline models in terms of accuracy and receiver operating characteristic curve (ROC), which proves that the urban spatio-temporal attribute of human mobility patterns plays a certain role in POI recommendation. It is worth noting that the performance of HMRec on the precision-recall curve (PR) is slightly lower than that of the DeepFM algorithm because the DeepFM model does not incorporate prior information on human mobility patterns and thus has a higher degree

of fitting to positive samples. Among all the baseline models, the performance of the Twin-tower model is relatively poor, which may be due to the fact that in the case of sparse samples, using a deep network for both users and points of interest can easily produce over-fitting results, resulting in the test set. The generalization ability is relatively poor. In contrast, NeuralCF achieves better results than the Twin-tower model due to its simplicity.

**Table 2.** Comparison of experimental results of the POI recommendation model.

| Model Algorithm | ACC | AUC (ROC) | AUC (PR) |
|---|---|---|---|
| **HMRec** | 0.8103 | 0.7331 | 0.3558 |
| **Wide&Deep** | 0.7978 | 0.7271 | 0.3499 |
| **DeepFM** | 0.8095 | 0.7253 | 0.3610 |
| **NeuralCF** | 0.7810 | 0.7078 | 0.3097 |
| **Twin Towers** | 0.6922 | 0.6097 | 0.1781 |
| **EmbeddingMLP** | 0.7966 | 0.7103 | 0.3302 |

By visualizing the ROC and PR curves (as shown in Figure 3), it can be seen that HMRec, Wide&Deep, and DeepFM are relatively close in terms of ROC curves. The curve of EmbeddingMLP has a steeper change in the middle, indicating that it does not perform well when predicting scores around 0.5. From the PR curve, it can be seen that the performance of HMRec, Wide&Deep, and DeepFM is still relatively stable, and Wide&Deep is not as good as HMRec and DeepFM after the recall rate increases. The performance of NeuralCF is very unstable, and the curve has large fluctuations. We can see that the HMRec represented by the red line achieves the best results.

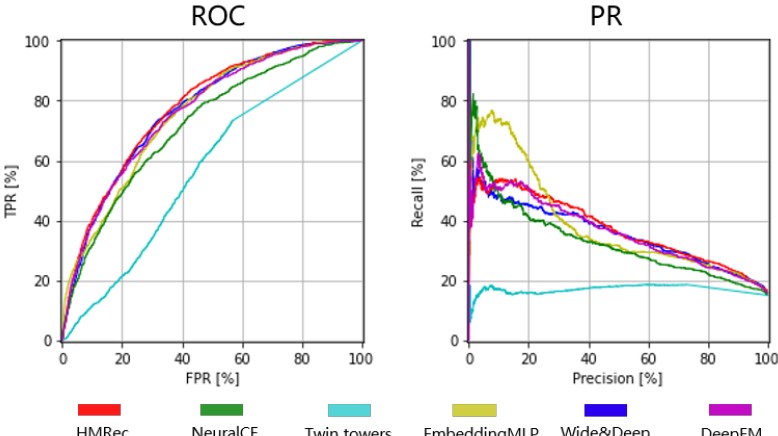

**Figure 3.** ROC and PR curve visualization of experimental results of recommendation algorithms. The left figure depicts draws the ROC curve of each model. The right figure depicts the PR curve of each model.

However, it must be acknowledged that our proposed method may have some drawbacks. The acquisition and analysis of crowd movement patterns may raise concerns regarding personal privacy and data protection. Although crowd movement patterns can provide useful information about user behavior, the sparsity of data may impact the effectiveness of the model as users typically visit only a small fraction of the overall set of locations. This can result in inaccurate predictions for certain locations or user behaviors.

## 5. Conclusions

At present, the existing research work on POI recommendation algorithms focuses on the mining of the interaction history between users and POIs and the research on users' social relations, ignoring the urban scene where POI recommendation is located and the objective existence of the prior knowledge of human mobility patterns. To this end, this

paper proposes a human mobility pattern extraction framework, HMPE, and designs an end-to-end pre-training process using graph neural networks and attention mechanisms. This paper proposes the HMRec POI recommendation algorithm to incorporate prior knowledge of human mobility patterns into POI recommendation. Experiments verify the effectiveness of human mobility patterns in POI recommendation.

In the future, we will attempt to mitigate the issue of data sparsity by employing data imputation techniques, and we will explore the design of multiple related downstream tasks to simultaneously obtain spatio-temporal prior knowledge from multiple tasks, aiming to comprehensively improve the effectiveness of POI recommendation.

**Author Contributions:** Conceptualization and methodology, J.S.; data curation and formal analysis, J.S. and Q.Y.; experiments and analysis, J.S., Q.Y., H.G. and B.W.; investigation, H.G.; validation and visualization, Q.Y., H.G. and B.W.; writing—original draft preparation, J.S., Q.Y. and H.G.; writing—review and editing, X.K. and Q.Y.; resources and supervision, X.K.; funding acquisition, X.K. and B.W. All authors have read and agreed to the published version of the manuscript.

**Funding:** This research was partially supported by the National Natural Science Foundation of China (62072409), Zhejiang Provincial Natural Science Foundation (LR21F020003), Program, for improving the Scientific Reasearch Ability of Youth Teachers of Inner Mongolia Agricultural University (RZ2200001860) and Major Science and Technology Projects of Inner Mongolia Autonomous Region (2020ZD0004).

**Institutional Review Board Statement:** Not applicable.

**Informed Consent Statement:** Not applicable.

**Data Availability Statement:** Not applicable.

**Conflicts of Interest:** The authors declare no conflict of interest.

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
