# Peer review of "Exploring Prior Knowledge from Human Mobility Patterns for POI Recommendation"

_applsci, doi:10.3390/app13116495_

Round 1

Reviewer 1 Report

1.I did not think using POI recommendation algorithm based on human mobility pattern  is a key contribution Authors need to establish clear superiority of their new methodology through comprehensive comparison results with very recent algorithms. The performance advantage is not significant. The contribution of this paper is marginal. Thus, the paper in the current form is not suitable for publication in this journal.

2. Author can refer to some latest related works from reputed journals like IEEE/ACM Transactions, Elsevier, Inderscience, Springer, Taylor & Francis, etc.

3. The authors seem to disregard or neglect some important results that have been recently achieved in this specific field.

3. Why Author not consider  Spatio-Temporal Gated Network for Next POI Recommendation,Justify?

4. Author has not point out the Time Complexity for the proposed model?

5. whether proposed Algorithm  can detect communities with high accuracy and stability.

6. Whether  proposed algorithm is validated by various networks and compared with several representative algorithms?

7.  Clarity of the figures can be improved Section 4. Experimental Results and Analysis.

Reviewer 2 Report

Please show the results quantitatively with a statistical sound in the Abstract. Then, it needs to provide more detail on the specific datasets or scenarios used in the experimental evaluation of the proposed algorithm. It would be helpful for readers to understand the context and scope of the experiments and how the proposed algorithm performs in different scenarios or with different types of data. The Abstract provides a general overview of the HMPE framework and the incorporation of prior knowledge of human mobility patterns into the model. It could be more explicit in outlining the specific steps and techniques used in the algorithm. The abstract mentions the limitations of current POI recommendation methods and the benefits of incorporating human mobility patterns. It does not discuss the potential drawbacks or challenges of the proposed approach. It impresses a one-sided view of the proposed algorithm and its potential impact in the field.

In the introduction, it is that it lacks clarity and organization. The author must discuss the importance of smart tourism in smart cities to introduce the concept of POI recommendation algorithms before providing a clear transition between the two topics. Additionally, the paper touches on various technologies, such as deep learning, IoT, and LBSNs, needing to fully explain their relevance to POI recommendation and human mobility pattern discovery. The reviewer had difficulties to follow the logical flow of the paper and understanding how the various concepts relate to each other. Furthermore, the paper could benefit from a more concise and focused research question that clearly states the problem being addressed and the specific contribution of the proposed method.

This paper needs a thorough discussion of the limitations of the study. While the authors compare their results with other baseline models and provide detailed experimental results, they do not discuss potential sources of error or limitations of their approach.

The conclusion section needs to provide a clear direction for future research. The author acknowledges that some problems still need to be solved, such as the sparsity problem of check-in data and the design of downstream tasks in different scenarios. However, the author offers no specific suggestions or recommendations for addressing these issues. Providing concrete suggestions for future study would help to advance the field further and build upon the work done in this paper. Additionally, the author could have discussed the potential limitations of their proposed approach and how future work could address them.

The reviewer only inspects the paper in general. English is generally clear and concise, with proper grammar and sentence structure. The paper uses technical language and terms related to computer science, but they are appropriately defined for readers who may need to become more familiar with them.

Reviewer 3 Report

1. The English writing style of this paper needs to be improved. For example,  "In order to efficiently mine human mobility pattern,..."  (line 43) I don't know whether the usage of the verb "mine"  is appropriate or not. Maybe the authors need to change another term.

2. From line 58 to line 64, the writing style is just like the direct translation from Chinese language. It needs to be improved.

3. On page 8, What is the difference between HMRec algorithm and HMPE algorithm ? The authors need to explain clearly.

4. What is the connection between the "Results and Discussion" and "Conclusion"?  The logical connection for both parts needs to be improved.

The quality English language writing needs to be improved.

Round 2

Reviewer 1 Report

Section 4 Table 1 Author may represent in the form of chart representation